# Considering the observation and illumination angular configuration for an improved detection and quantification of methane emissions

Javier Gorroño[1], Zhipeng Pei[2], Adriana Valverde[1], and Luis Guanter[1,3]

[1]Research Institute of Water and Environmental Engineering (IIAMA), Universitat Politècnica de València, València, Spain
[2]State Key Laboratory of Information Engineering in Surveying, Mapping and Remote Sensing, Wuhan University, Wuhan 430079, China
[3]Environmental Defense Fund, Reguliersgracht 79, 1017 LN Amsterdam, the Netherlands

**Correspondence:** Javier Gorroño (jagorvie@upv.es)

**Abstract.** A growing constellation of methane-sensitive instruments from space is making it possible to monitor methane point-source emissions from different industries and activities. Advanced plume simulation methods are becoming key to better understanding the uncertainties in the plume detection and quantification processes. This paper describes how observation and illumination angular configuration affects satellite-derived methane enhancement fields and, more critically, the calibration and uncertainty of semi-empirical emission rate estimation models, using simulated datasets. We review the mathematical expression of the retrieval and quantification methodologies to determine the uncertainty sources. We implement a method to simulate the apparent displacement of the plume when projected on the ground due to tilted illumination and observation (i.e., parallax effect). We apply this method to methane plumes generated from 3D spatial distributions of methane using the Weather and Research Forecasting Model in large-eddy simulation mode (WRF-LES). The results in the methane enhancement ($\Delta X_{\mathrm{CH_4}}$) maps show large spatial variations of the plumes with respect to a reference case with nadir observation and zenith illumination. In short, it suggests that the spatial distribution of the plumes is largely determined not only by the turbulence in the atmosphere but also by the acquisition's illumination and observation geometry. We tested the impact in the methane flux rate using the integrated mass enhancement (IME) method. Assuming nadir observation, these errors have a polar dependence with the solar angles that for our training dataset at $20 \times 20\,\mathrm{m}^2$ reaches values as high as the 30% error for high sun zenith angles orthogonal to the direction of the plume. The errors in the IME method are explained by the changing plume area with angular projection on the ground. Furthermore, it also has an impact on the probability of plume detection (PoD). The PoD is significantly reduced in the orthogonal plane to the wind direction and increases in opposite directions to the wind where the plumes are compressed. The dependence of the PoD on the angles increases with high values of $U_{10}$ with differences of up to 4 times between angular configurations at high sun zenith angles. Our results illustrate the importance of considering acquisition and observation geometry when analysing plume maps with a spatial sampling of $20\,\mathrm{m}$ or better.

## 1 Introduction

Anthropogenic methane emissions are responsible for about a third of global warming since the preindustrial period (Shukla et al., 2022). The oil and gas industry accounts for around 25% of anthropogenic emissions, and the mitigation of these types

of emissions has been found to be cost-effective (Ocko et al., 2021). In recent years, a growing constellation of space-based methane-sensitive instruments is capable of detecting and monitoring methane emissions from the oil and gas sector, but also from other sectors, such as waste management (Jacob et al., 2022).

The detection of methane plumes typically requires high spatial resolution (typically below 100 m) observations capable of defining the plume shape and attributing this to an emission source. Most of the retrieval of methane plumes is done using the spectral region around the Short-Wave InfraRed (SWIR) where methane absorption occurs (Cusworth et al., 2019; Jongaramrungruang et al., 2021).

Retrieval of methane enhancement $\Delta X_{\mathrm{CH_4}}$ from satellite observations requires the definition of a concentration throughout the column per pixel. The definition of this *column* is affected by the angles of observation and illumination following a slant path. Similarly to buildings or clouds in a scene, methane clouds suffer a displacement from the nadir because of the relatively small pixel of satellites with high spatial resolution (Wang et al., 2011). In that scenario, the downward and upward paths through a gas cloud might reach different pixels because of the parallax effect. We define the parallax of a methane plume as the horizontal displacement or shift between the apparent location of the plume from the satellite and solar perspective relative to the actual location of the scene projected vertically on the Earth's surface.

For any non-nadir viewing and/or non-zenith illumination, the resulting $\Delta X_{\mathrm{CH_4}}$ map will be affected by the parallax displacement. The effect will be diverse depending on the angular configuration, but it is clearly visible on the map when the illumination and satellite view are in the orthogonal plane to the plume direction and in opposite directions. In that case, the $\Delta X_{\mathrm{CH_4}}$ map might contain a *double plume*. For example, these can be found in airborne sensors such as AVIRIS in Borchardt et al. (2021) and Meier et al. (2026) or the WorldView-3 (WV3) satellite in Sánchez-García et al. (2022).

In the IME method, the dependence of the methane plume transmittance with the angle is accounted for with a dependence of the transmission vs. concentration considering the geometric air mass factor ($AMF_g$)(Jongaramrungruang et al., 2019; Varon et al., 2018). However, the slant path also has an implication on the shape of the plume, which, in turn, determines the relationship between $\Delta X_{\mathrm{CH_4}}$ and the area of the plume. Thus, the angular configuration has implications for both the $\Delta X_{\mathrm{CH_4}}$ map and the quantified flux rate.

In the literature, we find recent work in Eastwood et al. (2025) in which parallax has been applied in a retrieval method to plumes captured by the airborne visible/infrared imaging spectrometer (AVIRIS). A plume rise model was used to predict the height of the plume and the corresponding parallax motion.

This paper is devoted to the study in depth of the impact of the angular configuration in the detection and quantification methodologies of methane plumes. The analysis of $\Delta X_{\mathrm{CH_4}}$ maps is used to illustrate physical mechanisms, whereas the quantitative assessment focusses on emission flux calibration, the uncertainty, and the probability of detection. We work with vertically-resolved simulations of the volume mixing ratio (VMR) generated with the Weather and Research Forecasting Model in large-eddy simulation mode (WRF-LES). This is useful to understand the vertical distribution at the pixel level and, through realistic plumes, to understand the relative error patterns in the $\Delta X_{\mathrm{CH_4}}$ maps and quantification.

In Section 2, we propose a methodology to analyse the impact of the angular configuration on the maps of $\Delta X_{\mathrm{CH_4}}$ and flux rate quantification. First, we will introduce the concept and mathematical framework in Subsection 2.1 with Subsection

2.2 devoted to the definition of a methodology to include the angular configuration in the simulated plumes. Section 3 applies these concepts and methodologies to the $\Delta X_{\mathrm{CH_4}}$ maps (see Subsection 3.1), flux rate quantification (see Subsection 3.2) and the probability of detection (PoD) of methane plumes (see Subsection 3.3).

The observation and illumination angular configuration is a key element in closing the gap between models/simulations and satellite measurements. The examples proposed here must be used as illustrations and ultimately apply to the specific user case. Here, we discuss a few examples, but this concept could equally apply to other areas such as the development of training datasets for automated detection/quantification models or the design of methane-controlled releases. We have released the code and simulated data in Gorroño (2026) and Gorroño et al. (2026) to facilitate the study and adaptation to other cases.

## 2 Methodology to estimate the impact of the observation and illumination geometry on methane plume maps

### 2.1 The impact of angular configuration in point source emission quantification

The retrieval of methane enhancement $\Delta X_{\mathrm{CH_4}}$ from space observations is the first step to obtain an estimate of methane emissions. The column-averaged dry molar mixing ratio $X_{\mathrm{CH_4}}$ represents the concentration of methane molecules compared to the total volume of dry air in the atmospheric column. The enhancement $\Delta$ refers to the increase in methane from a reference background methane.

The $\Delta X_{\mathrm{CH_4}}$ map can be obtained with a physics-based algorithm such as the iterative maximum a posteriori differential optical absorption spectroscopy (IMAP-DOAS) algorithm (Frankenberg et al., 2005; Thorpe et al., 2014; Borchardt et al., 2021; Butz et al., 2011; O'Dell et al., 2018). The retrieval isolates the local features from gas absorption from those features that are less prone to rapid variations, such as surface reflectance and atmospheric scattering. More recently, the matched-filter (MF) method has been widely implemented in different satellite missions such as EnMAP, PRISMA, or EMIT (Thompson et al., 2016; Guanter et al., 2021; Irakulis-Loitxate et al., 2021). The matched-filter method fits the measured spectrum to a reference spectrum. The reference is created from spectra in the background observations in the image. The methane absorption spectrum is convolved with the instrument spectral line shape and then multiplied with reference spectrum, providing the target signature. The matched filter fits each spectrum using the reference spectrum and the target signature, providing a pixel-specific enhancement. (Thompson et al., 2015; Foote et al., 2020). In the case of band imagers / multispectral imagers such as Copernicus Sentinel-2 and Landsat-8/9, both a temporal and a spectral ratio of the bands can be used as a proxy of methane enhancement transmission (Varon et al., 2020).

The different retrieval methodologies must associate the spectral radiance / reflectance measurements with an atmospheric transmission of the methane plume $t_{\Delta X_{\mathrm{CH_4}}}$ and, in turn, relate this to the methane enhancement $\Delta X_{\mathrm{CH4}}$. Mathematically, this can be expressed as follows:

$$t_{\Delta X_{\mathrm{CH_4}}}(\lambda) = e^{-\mathrm{AMF_g} \cdot \sigma_{\mathrm{CH_4}} \cdot \Delta X_{\mathrm{CH_4}}} \tag{1}$$

where $\sigma_{\mathrm{CH_4}}$ refers to the methane absorption cross section and $AMF_{\mathrm{g}}$ refers to the geometric air–mass factor.

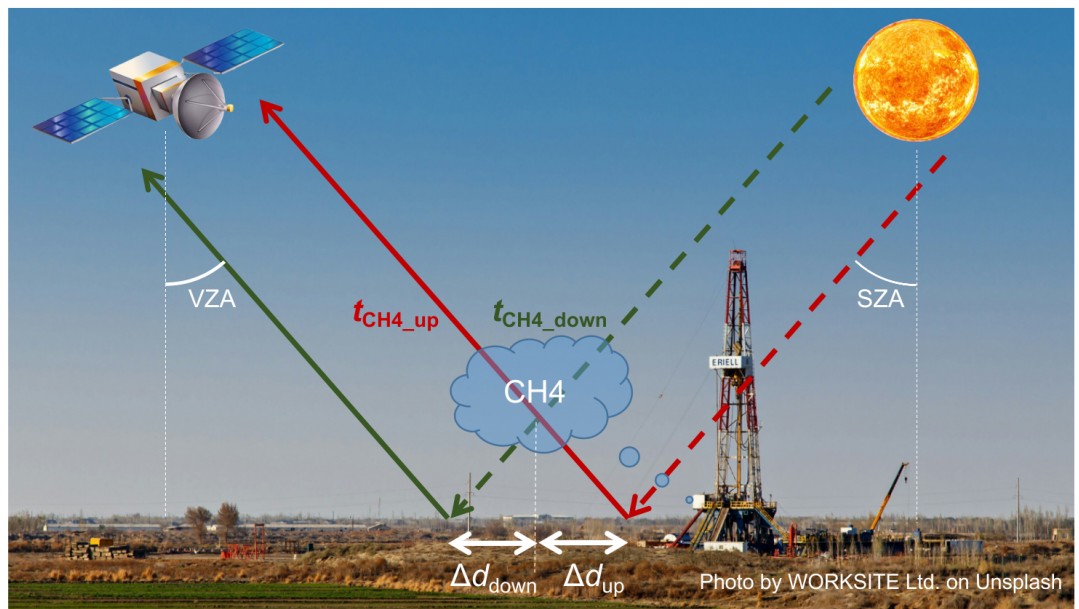

**Figure 1.** Diagram describing the angular dependency of the simulation and the spatial shift $\Delta d_{\mathrm{up}}$ and $\Delta d_{\mathrm{down}}$ produced by the parallax.

The $AMF_{\mathrm{g}}$ can be defined as the ratio between the slant column density and the vertical column density. The optical path depends on the illumination and viewing geometry and the geometric optical path can be written as:

$$AMF_{\mathrm{g}} = 1/cos(SZA) + 1/cos(VZA) \tag{2}$$

where $VZA$ and $SZA$ refer to the viewing zenith angle and the solar zenith angle, respectively.

    This formulation must assume a non-scattering atmosphere and horizontal homogeneity of the gases. The former is generally

valid in the SWIR region, where the scattering is almost negligible. The horizontal homogeneity is also valid in some cases, such as for coarse-resolution satellites (e.g. TROPOMI), where horizontal variability is averaged over a large pixel size or gas diffusion in the atmosphere is sufficiently uniform. Furthermore, Equation 2 also assumes a plane-parallel atmosphere. That might be invalid for very high values of SZA or VZA, but does not typically occur for nadir-viewing instruments.

    For satellites that target point-source emissions with a high spatial resolution, the relatively small pixel cannot assume a

homogeneous path within the pixel. In that scenario, the downward and upward paths through a gas cloud might reach different pixels because of the parallax effect.

    The direct component of light interacts twice with the methane plume, and no parallax will be achieved only when illumination is zenith (SZA=0) and observation is nadir (VZA=0). This is rarely achieved since most illumination scenarios are far from a zenith illumination. Figure 1 illustrates this scenario with a simplified 2D representation.

As mentioned above, the direct component of light interacts twice with the methane concentration. First, before reaching the surface (*downwelling path* is a dashed green line), and again, once it reflects the surface (*upwelling path* is a continuous red

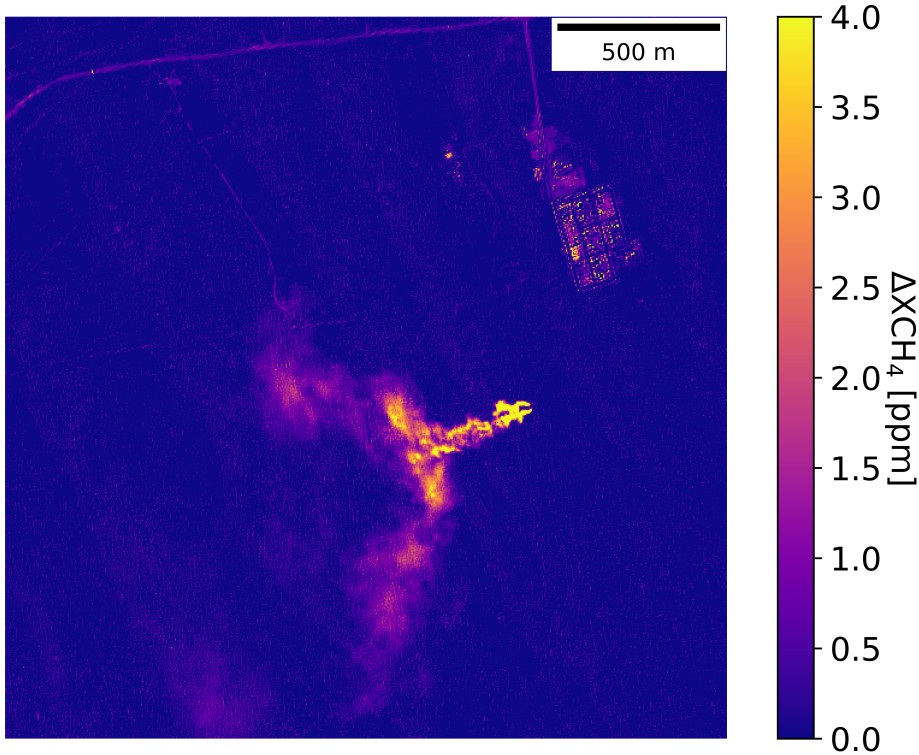

**Figure 2.** $\Delta X_{\mathrm{CH_4}}$ map over a region in of Turkmenistan South; Korpezhe O&G field on 2021/03/29. The plume is located in 38.494°N, 54.198°E. VZA is approximately 27.5° and SZA 37.5°. The azimuth illumination and observations are 154.5° and 33.1° respectively. Reprocessed from Sánchez-García et al. (2022).

line). This results in two different transmittances for the downwelling $t^{\mathrm{down}}_{\Delta X_{\mathrm{CH_4}}}$ and upwelling path $t^{\mathrm{up}}_{\Delta X_{\mathrm{CH_4}}}$. There is a parallax displacement for the upwelling and downwelling paths, labelled $\Delta d_{\mathrm{up}}$ and $\Delta d_{\mathrm{down}}$, respectively.

This has a direct impact on the concentration map because at pixel level, the contributions might differ from different paths
and plume areas. Figure 2 shows a double plume as presented in a case study by Sánchez-García et al. (2022).

It follows a similar angular configuration illustrated in Figure 1. In this case, the VZA is approximately 27.5° and the SZA 37.5°. In addition, the plume runs towards the south-west with an azimuth illumination and observations of 154.5° and 33.1°, respectively. That is, the double plume structure is a consequence of a similar viewing and sun zenith angles in almost different directions in the azimuth range and near the orthogonal plane to the plume direction.

In this specific angular configuration, the ground projection results in two separated and nearly symmetric images. For its quantification, $AMF_{\mathrm{g}}$ is overestimated because only the downward or upward paths must be considered for each pixel. Then, the impact on $\Delta X_{\mathrm{CH_4}}$ is inversely proportional to the $AMF_{\mathrm{g}}$ attending to equation 2. For example, if we had equal values of

SZA and VZA, $AMF_g$ would be double for each pixel, leading to half the value of all pixels on the $\Delta X_{CH_4}$ map (assuming that there is no noise and artefacts).

We have selected this specific angular configuration for illustrative purposes. However, other angular configurations lead to different parallax displacements. In general, this double-plume example must be considered to be the most extreme case as compared to an ideal zenith illumination and nadir observation. The more general scenario will include a partial overlap between the two paths. We include an animation accessible in Valverde and Gorroño (2025) that displays a mock-up of a methane plume with ground projections for both illumination and observation conditions. We set the observation to nadir and

the illumination changes to different azimuth values, creating different projections from an elongated plume to a contracted one including a double plume in between. Quantified $\Delta X_{CH_4}$ maps at different angular configurations will also be displayed in Subsection 3.1.

Once the $\Delta X_{CH_4}$ map has been obtained, the processing continues with the detection and quantification of methane plumes. Current methods rely on the mass-balance concept and associate methane concentration enhancements with single point source

emission rates.

The most widely used method for the quantification of plume emissions is the integrated mass enhancement (IME) method, which was first introduced by Frankenberg et al. (2016) for aircraft measurements. The method is applied to satellite data in Varon et al. (2018) where they introduce the terminology of effective wind speed and plume length. For simplification reasons, the latter method is selected here to describe the uncertainty of reported flux rates with concepts readily applicable to variations

and other methods such as the cross-sectional flux (CSF) method (Krings et al., 2011).

The method estimates the methane flux rate as the total mass enhancement in the plume scaled with a coefficient dependent on the wind speed. The $IME$ in kg is defined as the total excess mass of methane contained in the plume,

$$\text{IME} = k \sum_{j=1}^{n_p} \Delta_{XCH_4}(j), \tag{3}$$

where $n_p$ is the number of pixels in the plume and $k$ is a scaling factor that converts the total of the pixel-wise methane

concentration values in ppb to kg by assuming Avogadro's law, the molar mass of methane (0.01604 kg/mol), an atmospheric column height of 8000 m and taking into account the pixel size (e.g. $k = 5.155 \cdot 10^{-3}$ kg/ppb for 30 m pixel).

In relation to the previous example of an ideal double plume, the underestimate in $\Delta X_{CH_4}$ map would be cancelled in $IME$ since the pixels would be accounted for twice taking into account equation 3. Therefore, in real terms, we can expect a robust $IME$ against angular configurations because the errors in $AMF_g$ and $n_p$ tend to cancel each other out.

This total mass enhancement is scaled to obtain the flux rate $Q$ as follows:

$$Q = \frac{U_{eff} \cdot \text{IME}}{L}, \tag{4}$$

where $U_{eff}$ is an effective wind speed and $L$ the plume length scale in m, which is typically approximated by the square root of the plume mask area, following the standard IME formulation of Varon et al. (2018).

Regarding $U_{eff}$, this term is intended to account for turbulent diffusion of the methane flux, and also for the fraction of the

plume which is not detected due to retrieval noise (Varon et al., 2018). $U_{eff}$ can be expressed as a linear or logarithmic function

of the measurable 10-m wind speed $U_{10}$ available from the reanalysis products. This function can be defined with modelled plume simulations thereby accounting for the corresponding spatial sampling and retrieval noise of each specific instrument.

The angular configuration also has an impact on the terms $L$ and $U_{\text{eff}}$. However, its impact is based on the definition of both $L$ in equation 4 and $L_{\text{ref}}$ during $U_{\text{eff}}$ calibration. In that process, each one of the samples defines the regression between $U_{\text{eff}}$ and $U_{10}$ and can be defined as:

$$U_{eff(i)} = \frac{Q_{\text{ref}} \cdot L_{\text{ref}(i)}}{IME_{\text{ref}(i)}} \tag{5}$$

where $i$ refers to the index of the simulated plume.

Taking into account both equations 4 and 5, the impact on emission quantification is based on systematic differences in factor $L_{\text{ref}(i)}/L$. The evaluation of this factor becomes more complex since, as mentioned above, the upwelling and downwelling paths partially overlap. In addition, the absence of 3D information for real plumes does not simplify understanding of the directional impact. Thus, in the next subsection 2.2, we will look at the impact of the angular configuration supported with atmospheric transport simulations. We will be able to compare a plume 3D projection on nadir (as typically considered) against other angular configuration. In short, we will produce $\Delta X_{\text{CH}_4}$ maps and $U_{\text{eff}}$ calibration factors that recreate conditions closer to retrieval, by minimising the mismatch between $L_{\text{ref}(i)}$ and $L$, eventually leading to lower uncertainty.

## 2.2 Simulation of the angular configuration impact on the observation of methane plumes

We used WRF-LES to study this angular impact. Realistic methane plumes represent a three-dimensional mass distribution in the atmospheric boundary layer over time, similar to the approach in Varon et al. (2018). The simulation was carried out at 20 $\times$ 20 m$^2$ spatial resolution with different geostrophic wind fields between 2-10 m s$^{-1}$ and a sensible heat flux $H = 150\,\text{W}\,\text{m}^{-2}$. The vertical resolution was also set to 20 m. For each geostrophic wind, two 1-hour simulation are obtained. We discard the first 30 minutes of each one of the two simulations and retrieve a 3-D methane distribution every 60 seconds, leading to a total of 60 snapshots per geostrophic wind. Based on Ouerghi et al. (2025), the selected separation time between samples is a suboptimal case selected as a trade-off between correlation between samples and sample size. The resulting 3-D methane distributions can be vertically integrated to obtain a 2D mass field for column mass enhancement $\Delta\Omega$ in kg m$^{-2}$.

These maps are further converted into $\Delta X_{\text{CH}_4}$ maps as follows:

$$\Delta\Omega = \frac{M_{\text{CH}_4}}{M_{\text{a}}}\Omega_{\text{a}}\Delta X_{\text{CH}_4} \tag{6}$$

where $M_{\text{a}}$ and $M_{\text{CH}_4}$ refer to the molar mass of dry air and methane, respectively [kg mol$^{-1}$], and $\Omega_{\text{a}}$ refers to the dry air column [kg m$^{-2}$].

These $\Delta X_{\text{CH}_4}$ maps are associated with a reference flux rate $Q_{\text{ref}}$ that can be obtained from the initial snapshots of the simulation. Then, this reference flux rate can be linearly scaled to the desired level following Guanter et al. (2021) and Sánchez-García et al. (2022).

The vertical integration of the layers for a non-nadir observation and/or non-zenith illumination (i.e. different from SZA = VZA = 0°) requires a specific modelling of the parallax displacement as illustrated in Figure 1. This apparent displacement of the plume

is defined as $\Delta d$ in the voxel $i, j, z$, which can be expressed as:

$$\Delta d_{i,j,z} = h_{i,j,z} \cdot tan(ZA) \tag{7}$$

where $h_{i,j,z}$ expresses the altitude of the voxel that here is calculated based on the layer index and vertical spatial resolution. $ZA$ refers to the viewing or sun zenith angle in radians.

We can further decompose this shift into spatial components for the x and y dimensions as follows:

$$\Delta d_x = \Delta d_{i,j,z} \cdot cos(AA)$$
$$\Delta d_y = \Delta d_{i,j,z} \cdot sin(AA) \tag{8}$$

where $AA$ refers to the azimuth of the viewing or solar angles.

The code in Gorroño (2026) collapses the vertical dimension, considering $\Delta d_x$ and $\Delta d_y$ in each vertical layer. Because we have two different paths (downwelling and upwelling; see Figure 1), it results in two different $\Delta X_{CH_4}$ maps ($\Delta X_{CH_4}^{up}$ and $\Delta X_{CH_4}^{down}$) with their corresponding spatial shifts at the pixel level. For the first map, we consider the viewing angles and the former map considers the solar angles.

Taking into account the two paths, we can rewrite the plume transmittance in equation 1 as a combination of upward and downward transmission ($t_{\Delta X_{CH_4}}^{up}$ and $t_{\Delta X_{CH_4}}^{down}$) and the geometric air mass factors ($AMF_{up}$ and $AMF_{down}$):

$$t_{\Delta X_{CH_4}} = t_{\Delta X_{CH_4}}^{up} \cdot t_{\Delta X_{CH_4}}^{down} = e^{-(AMF_{down} + AMF_{up}) \cdot \sigma_{CH_4} \cdot \Delta X_{CH_4}} \tag{9}$$

with $AMF_{up}$ and $AMF_{down}$ representing the split of the geometric air mass factor as follows:

$$AMF_g = AMF_{down} + AMF_{up} = 1/cos(SZA) + 1/cos(VZA) \tag{10}$$

and $t_{\Delta X_{CH_4}}^{up}$ and $t_{\Delta X_{CH_4}}^{down}$ defined as:

$$t_{\Delta X_{CH_4}}^{up} = e^{-AMF_{up} \cdot \sigma_{CH_4} \cdot \Delta X_{CH_4}^{up}}$$
$$t_{\Delta X_{CH_4}}^{down} = e^{-AMF_{down} \cdot \sigma_{CH_4} \cdot \Delta X_{CH_4}^{down}} \tag{11}$$

Considering equation 11 and applying the rule of product of powers in equation 11, we can establish a direct link between each of the enhancement maps and obtain the combined methane enhancement $\Delta X_{CH_4}$ as the weighted average of the two enhancement maps $\Delta X_{CH_4}^{up}$ and $\Delta X_{CH_4}^{down}$:

$$AMF_{up} \cdot \Delta X_{CH_4}^{up} + AMF_{down} \cdot \Delta X_{CH_4}^{down} = (AMF_{up} + AMF_{down}) \cdot \Delta X_{CH_4} \tag{12}$$

The resulting $\Delta X_{CH_4}$ maps incorporate the shape distortion similarly to the satellite observations. Equation 10 highlights the explicit dependence of the geometric air mass factor on SZA and VZA, which are systematically varied in this study to assess their impact on methane flux quantification. In the following subsections 3.1 and 3.2 separately exemplify, with results, the impact on $\Delta X_{CH_4}$ and $U_{eff}$, respectively.

## 3  Results

### 3.1  Impact on methane enhancement maps

In this subsection, the impact on $\Delta X_{CH_4}$ maps is analysed to illustrate changes in plume shape, without attempting a quantitative retrieval performance assessment. From the simulations described in subsection 2.2 we have selected a specific snapshot and a geostrophic wind of 2 m/s. The direction of the geostrophic wind is set towards the right of the image. In both cases, we set a flux rate of Q = 3 t/h. However, this value is not highly relevant since the angular changes are expected to be proportional. Two different cases have been defined:

- (SZA 60°, VZA 0°). VZA is fixed to the nadir, since many satellites point to the nadir (irrespectively of the swath dimensions). The SZA value of 60° represents a typical mid-latitude winter scenario.

- (SZA 40°, VZA 40°).The viewing azimuth angle is orthogonal to the direction of the geostrophic wind. This off-nadir angle is considerably large, but can be reached by many agile systems such as GHGSat (Jervis et al., 2021). It is also reached by large-swath instruments, although typically displaying larger spatial resolutions than 20 m. The SZA is reduced to 40° to maximise the visual effect of a symmetric double plume, while at the same time this solar angle represents a standard value at many latitudes.

Figure 3 includes the $\Delta X_{CH_4}$ maps for the projection plan and elevations in the directions across and along the plume. The maps are calculated collapsing the across and along the plume directions rather than the vertical one. Figure 4 includes the $\Delta X_{CH_4}$ maps for the relative azimuth angles (RAA) values of 0°, 90°, -90° and 180 ° and for the two cases defined above. The central panel represents the ideal zenith illumination and nadir observation as in Figure 3a. The RAA is defined as the relative angle between the solar azimuth angle and the direction of the geostrophic wind.

The panels in Figure 4 clearly demonstrate how a plume can vary significantly from a nadir angle assumption. The plumes on the right (RAA 0°) appear elongated, whereas the plumes of RAA 270° show plume compression with generally higher values of $\Delta X_{CH_4}$. The compression and elongation effect is more extreme for the case (SZA 60°, VZA 0°). This can be explained because the upwelling path is not modified from the ideal zenith illumination and nadir observation, but the downwelling path $\Delta X_{CH_4}^{down}$ does.

For RAA 90°, it results in a displacement to the right of the downward path, and the opposite occurs for -90°. For the case (SZA 60°, VZA 0°) it shows a fainter replica of the central reference panel due to the upwelling transmittance at the nadir combined with a replica distorted by the angle effect. For the case (SZA 40°, VZA 40°), the plume at RAA 90° represents a distorted plume from the central reference panel, but with higher concentration values because the light has the same upwelling and downwelling paths. Finally, for the same case (SZA 40°, VZA 40°), the plume at RAA -90° represents a nearly symmetric double plume very similar in shape and angular configuration to the WV3 plume presented in Figure 2.

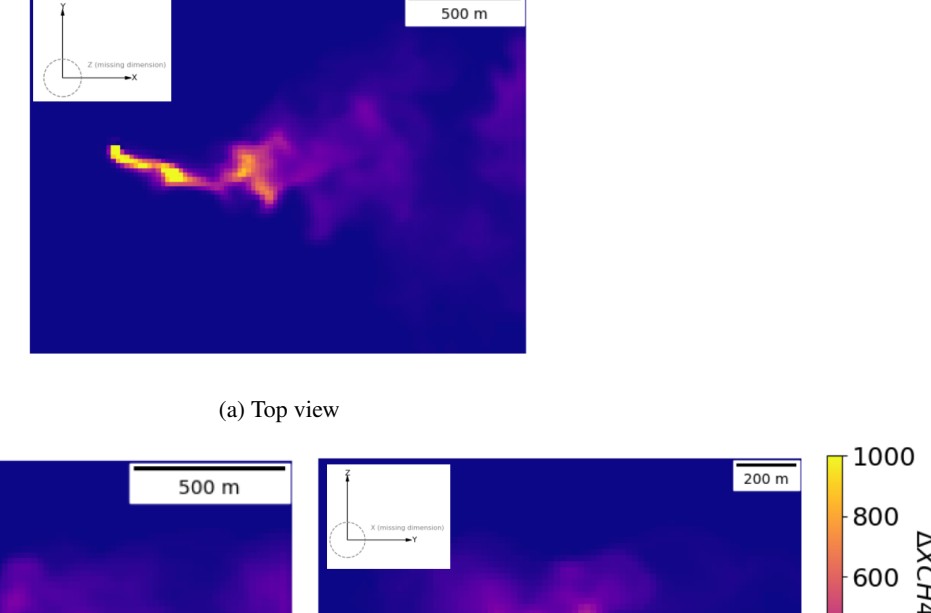

(a) Top view

(b) Along plume elevation

(c) Across plume elevation

**Figure 3.** $\Delta X_{\text{CH4}}$ top view projection (3a), elevation in the along direction (3b) and the across-track direction (3c) of the selected plume. A small axis glyph is included in each panel to indicate the along-plume (x), cross-wind (y), and vertical (z) directions.

## 3.2 Impact on methane quantification

Subsection 3.1 illustrates the large impact of the angular configuration on methane concentration maps. These maps can also be used to train a $U_{\text{eff}}$ model considering the angular configuration for each snapshot. We have set training based on two separate 1-hour simulations with 11 geostrophic winds from 2 to 10 m/s. For each case, we only consider, as a conservative criterion, the second half hour of the simulation. Since we collect snapshots every 60 seconds, we have a total of 660 samples.

    To illustrate the impact, we have selected a flux rate of 3 t/h and included 50 ppb noise. This could respond to the expected
noise in a hyperspectral mission such as EnMAP and EMIT (note that we consider that the spatial resolution is different). We apply filtering twice the noise to the enhancement images and define a detected plume when more than 10 consecutive pixels are detected. The simulated scene of $1.8 \times 1.2$ km might be slightly clipping the masks, but this effect is very limited. The $U_{\text{eff}}$-$U_{10}$ relationship is established with a Huber regression. Figure 5 includes the scatter plots for the $U_{\text{eff}}$-$U_{10}$ points and the

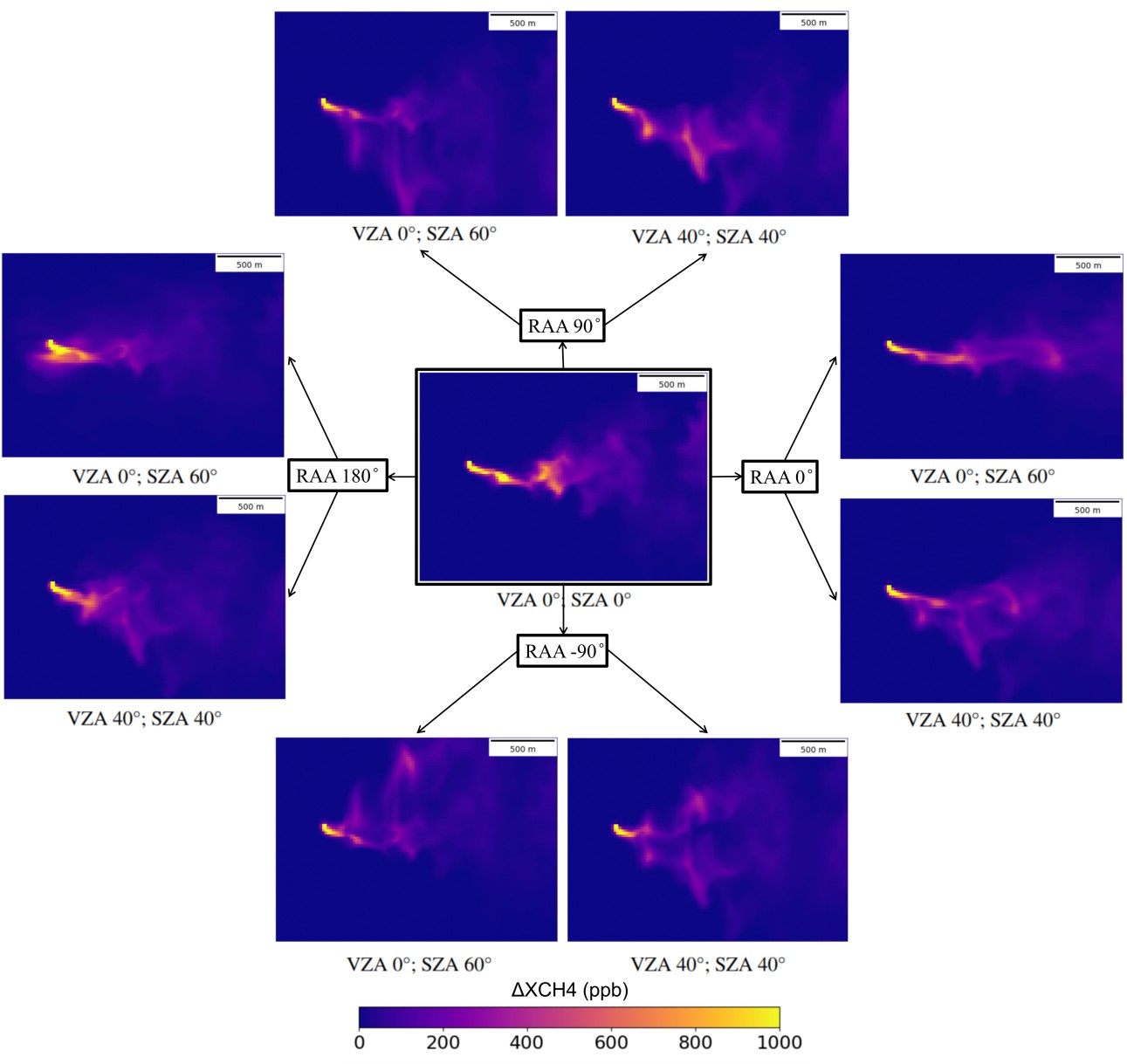

**Figure 4.** $\Delta X_{CH_4}$ map at different RAA values for a mid-latitude winter case (SZA 60°, VZA 0°) and a standard angular configuration of a pointing system (SZA 40°, VZA 40°). For all cases the geostrophic wind direction is eastwards RAA = 0°.

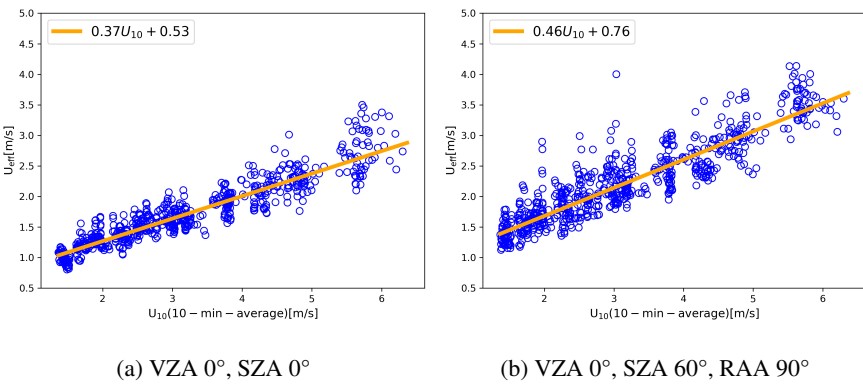

(a) VZA 0°, SZA 0°                        (b) VZA 0°, SZA 60°, RAA 90°

**Figure 5.** Scatter of $U_{\text{eff}(i)}$ samples against $U_{10(i)}$ for angles (5a) VZA 0°, SZA 0°and (5b) VZA 0°, SZA 60°, RAA 90°. The graph includes the fitted $U_{\text{eff}}$ equation.

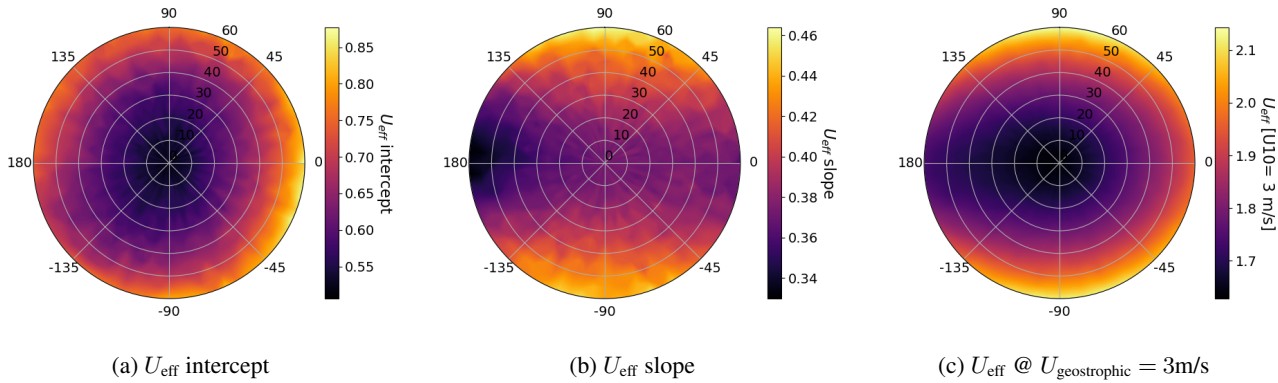

(a) $U_{\text{eff}}$ intercept           (b) $U_{\text{eff}}$ slope           (c) $U_{\text{eff}}$ @ $U_{\text{geostrophic}} = 3$m/s

**Figure 6.** (6a) $U_{\text{eff}}$ intercept, (6b) slope and (6c) reference value at a $U_{10}$ of 3 m/s considering Q=3 t/h and noise of 50 ppm. Polar plot represents values of sun angles for a viewing nadir. The angular axis represents the RAA angles and radial axis refers to the SZA angles.

fitted $U_{\text{eff}}$ for two cases. Figure 5a) represents the angular configuration of VZA 0°, SZA 0° and Figure 5b does it for VZA 0°,
SZA 60°, RAA 90°.

Both the intercept and slope increase for the non-nadir and non-zenith configuration as well as for the scattering of the points. A more general understanding is given in Figure 6 which better captures the effect on slope and intercept values in a polar plot with varying SZA/RAA values and considering a nadir view. It also includes the values of $U_{\text{eff}}$ in a reference $U_{10} = 3\,m/s$.

The intercept in Figure 6a increases in all cases as the SZA increases, and the slope in Figure 6b does it for all cases
except for case (SZA 60°, VZA 0°) and SAA 270°. If we consider equation 5, we observe that the samples of $U_{\text{eff}(i)}$ increase proportionally to the increase of $L_{\text{ref}(i)}$. The values of $L_{\text{ref}(i)}$ will generally be equal or greater than the plume length at the nadir. Therefore, it is expected that the value of $U_{\text{eff}(i)}$ increases.

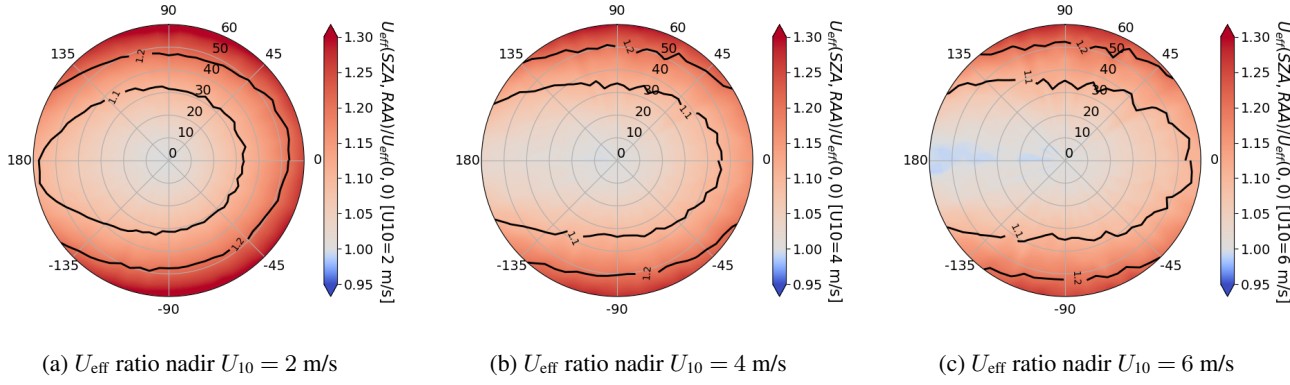

(a) $U_{eff}$ ratio nadir $U_{10} = 2$ m/s    (b) $U_{eff}$ ratio nadir $U_{10} = 4$ m/s    (c) $U_{eff}$ ratio nadir $U_{10} = 6$ m/s

**Figure 7.** $U_{eff}$ error ratio against nadir considering Q = 3 t/h and noise of 50 ppm. Polar plot represents values of sun angles for a viewing nadir. The angular axis represents the RAA angles and radial axis refers to the SZA angles.

Both the intercept and the slope show clear patterns. The intercept generally increases with SZA with a certain level of anisotropy. However, the slope is reduced for RAA values around 180° and increases in the orthogonal direction. This can be
explained due to the compression effect described in Subsection 3.1. The $U_{eff(i)}$ values for the $U_{10}$ range tend to increase and result in a higher intercept, whereas the slope is decreased because for higher winds the $U_{eff(i)}$ are comparable or even lower. It is a complex behaviour that can be explained since, at lower winds, the compression of the plume also results in a broadening of it whereas at larger winds, the plume tends to follow a less turbulent shape and the compression includes an overlap of the upwelling and downwelling transmittances.
The combined effect for a reference geostrophic wind of 3 m/s is shown in Figure 6c. It shows a clear pattern of $U_{eff}$ that increases towards the high SZA in the orthogonal plane to the direction of the plume. Figure 4 shows plumes that are more dispersed in RAA 90° and -90° due to the broadening of the plumes. This is translated into an increase in the ratio $L_{ref(i)}/L$ (see equation 5) and, consequently, in $U_{eff}$.

The value of $U_{eff}$ also slightly increases towards the same direction as the geostrophic wind and remains similar for the
opposite direction. This is produced by the effect of contraction and elongation of the plume described in Subsection 3.1.

Defining the values of $U_{eff}$ based on $\Delta X_{CH_4}$ maps from the vertical integration (SZA=0 and VZA=0) of WRF-LES model results in errors. Figure 7 normalises $U_{eff}$ against the vertical integration scenario (SZA=0 and VZA=0) to show the relative error in the polar coordinates for different $U_{10}$ winds.

The errors in Figure 7 show that there is a strong dependence on the azimuth and zenith configuration. The errors show
systematically larger values to the orthogonal direction of the wind mimicking Figure 6c. The errors range between 0 and 30%. If we consider an orbit in near-polar Sun synchronous orbit with a local mean equatorial crossing time around 10:30 (which could be S2 or Landsat missions), the errors for mid-latitude summer and equatorial cases will be contained within 10%. However, for winter mid-latitude scenarios, these errors will typically be in the 10% to 30% range.

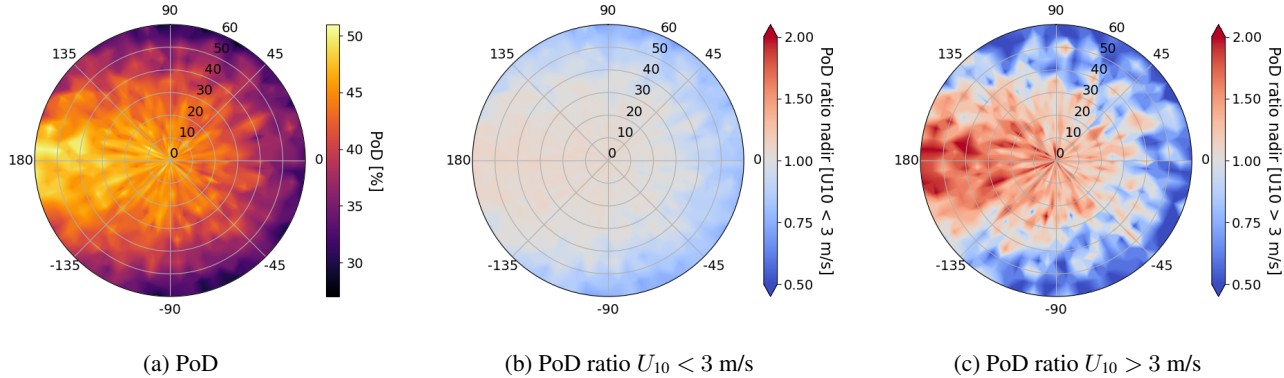

**Figure 8.** (8a)PoD considering a flux rate of 3 t/h and background noise of 200 ppm. (8b) and (8c)indicate the ratio of change of PoD from nadir for values of $U_{10}$ below and above 3 m/s, respectively. Polar plot represents values of sun angles for a viewing nadir. The angular axis represents the RAA angles and radial axis refers to the SZA angles. The dataset considered contains all the plumes described in Subsection 2.2.

We find residual values below 1 that are expected to occur for a RAA value close to 180°. As commented in Subsection 3.1, in those areas the plumes are compressed, and the plume length can be even below the nadir value. This explains why the errors tend to decrease as the wind values increase.

### 3.3   Impact on probability of detection

The results in Figures 6c and 7 show a clear pattern of errors in $U_{\text{eff}}$. As commented in Subsection 3.2, this is the consequence of changing the $L_{\text{ref}(i)}/L$ with different solar angles. The example set plumes with a flux rate of 3 t/h and a background noise of 50 ppb. This resulted in the detection of all the plumes in the training dataset. If we were not to detect all the plumes in the dataset, the logical assumption would be that this has an impact on the probability of detection (PoD). That is, the lower the plume length for a same flux rate, the higher chance to be detected and vice versa.

To test this hypothesis, we increase the noise to 300 ppb so that only a fraction of the dataset is detected (decreasing the flow rate would be analogous). We model a $U_{\text{eff}}$ for different angular configurations and calculate the number of plumes detected through the dataset as the PoD. The resulting PoD for a flux rate of 3 t/h is shown in Figure 8a.

The PoD shows a clear dependence on the angular configuration that ranges from 25 to 50%. The higher chances of detection occur for RAA= 180. This is the area where the plumes are compressed, as shown in Figure 4. The result indicates that compression, elongation, or broadening of the plumes results in sensible differences in the chances of detecting methane plumes.

We divide the results for values above or below $U_{10} = 3$ m/s in Figures 8b and 8c, respectively. These figures show the PoD ratio against the nadir case. It reports small changes in low wind. However, for larger winds, the PoD can be double from the nadir and becomes 4 times larger from peak-to-peak.

In general, it suggests that for large solar angles, the PoD is significantly reduced in the orthogonal plane to the wind direction. For solar angles in opposite directions to the wind, the PoD increases since the plumes are compressed, and higher values of $\Delta X_{CH_4}$ increase its detection chances. The dependence of the PoD on the angles increases with high values of $U_{10}$.

## 4  Discussion and conclusions

In the last decade, there has been a growing use of satellite instruments to detect and quantify methane point source emissions. In parallel, retrieval methodologies and automated detection/quantification methods are under constant improvement.

This paper explores the impact of the angular configuration in the observation of methane plumes. The plumes rise in the atmosphere as they are transported from the source. This leads to an apparent displacement from the nadir (i.e. parallax) for both the downward and upward radiation. Integration of parallax in detection and quantification methods could have a major impact on the detection and quantification models currently in use or under development.

Satellite observations only provide a 2D representation of the TOA radiance. However, the use of LES simulations defines a realistic 3D representation of the gas concentration in the atmosphere that here is used to estimate the impact on the retrieval and quantification methods. These voxels must be converted into a 2D representation of simulated methane enhancement considering the light path between the sun and the sensor. The ideal scenario requires integrating the simulations into a 3D radiative transfer code, for example, in Schwaerzel et al. (2020). This approach is complex and computationally expensive.

For the particular case of methane retrievals in the SWIR spectral range, the impact of atmospheric scattering is generally very small. In this scenario, assuming no atmospheric scattering and only accounting for the direct light component is a fair assumption. In addition, plane-parallel can be assumed since the plume boundary layer and extension of the plume are generally within kilometres.

We take these assumptions to propose a methodology to estimate the parallax impact on the plume projection. The different vertical layers are compressed by applying a specific shift for each of them. The method considers an independent shift for the downward and upward paths. The result is two enhancement maps ($\Delta X_{CH_4}^{up}$ and $\Delta X_{CH_4}^{down}$) that are merged into a single $\Delta X_{CH_4}$ map.

Figure 4 illustrates the impact of the angular configuration in the simulated $\Delta X_{CH_4}$ maps. The plume is elongated and compressed in the plane of the plume direction and, in the orthogonal plane to the plume direction, it results in a broadening of the plume with a *double plume* in extreme cases. We also noted that for similar angular configurations, the $\Delta X_{CH_4}$ map in Figure 3 closely matches the real *double plume* observation in Figure 2. In general, these results indicate that not only turbulence in the atmosphere is a good predictor of the 2D spatial distribution of the plumes, but the angular configuration might play a similar or even larger role.

The change in the $\Delta X_{CH_4}$ maps is translated into errors in the $U_{eff}$ calibration. Consequently, uncertainties in $\Delta X_{CH_4}$ are not evaluated independently, but through their propagation to flux quantification. Taking into account equation 5, the samples of $U_{eff(i)}$ increase proportionally to the increase of $L_{ref(i)}$. Changes in the values of $L_{ref(i)}$ are translated into changes in the value of $U_{eff(i)}$ and finally in the calibration of the flux rate. Figure 5 shows the translated maps from Figure 4 into a scatter of

$U_{\text{eff}(i)}$ values. These are represented in polar coordinates in Figures 6 and 7 assuming a nadir observation for the coefficients and errors, respectively. The results indicate a large dependence of the error with the solar angles. Considering a near-polar Sun synchronous orbit, the errors for mid-latitude summer and equatorial crossing will typically be below 10%. For mid-latitude winter scenarios, these errors will increase up to 30%.

For the IME implementation, it is feasible to create a correction of the $U_{\text{eff}}$ calibration for different angles of observation and illumination. Its application in operational scenarios requires the plume direction that can be extracted from the image itself, or the wind direction that can be used as a proxy value. An alternative consideration is to adjust the plume shape to a fixed angular configuration. However, it would require an inference of the 3D structure. There are options that include plume rise models as applied in Eastwood et al. (2025). However, this is a challenging approach when considering a per-pixel effect.

Changes in $\Delta X_{\text{CH}_4}$ and quantification due to angular configuration suggest that it can also impact the PoD. Figure 8a has modelled the PoD in polar coordinates and finds a drop from 50% to 25% for a large SZA in the orthogonal plane to the plume direction. Figures8b and 8c complement these findings, indicating larger variations as the wind increases. These findings are easily explained if we tie these variations to the compression, elongation, and broadening of plumes as presented in Figure 4.

     Explicitly considering angular effects naturally motivates a broader discussion of quantification methods and training strate-
gies. For example, it is possible that other methods, such as the Cross-Sectional Flux, are more robust against the parallax effect since no area is required. As mentioned above, we have released the code and simulated data in Gorroño (2026) and Gorroño et al. (2026) to facilitate the application to other scenarios. Without expanding into a methodological comparison, we also tested an alternative definition of the plume length scale $L$ based on the maximum plume extent after noise addition, rather than the square root of the plume mask area used throughout this study. This alternative formulation results in a redistribution
of the angular error patterns, with errors tending to concentrate preferentially in the lateral directions of the polar representation rather than symmetrically on either side. This is expected from a plume length definition less affected by plume shape broadening. Nevertheless, the overall magnitude of the errors and their systematic dependence on the observation and illumination geometry remain consistent. These results indicate that, while the specific definition of L influences the spatial expression of the errors, the dominant driver of the uncertainty is the angular configuration itself, reinforcing the main conclusions of this
work.

     The training dataset in the examples is sufficient to illustrate these angular effects but might require more cases and improvements. For example, a lower vertical resolution than the horizontal one might be a better design, since it can better capture the angular effects. This comes at the expense of larger processing and memory requirements. In general, modelling of more specific cases requires a more complex scenario that could be considered in future upgrades. For example, there could be a
dependency of the noise applied in relation to the surface bidirectional reflectance and atmospheric directionality. In addition, we could also model emissions that rise from a certain altitude or velocity rather than from the surface. In some scenarios, these emissions might originate from stack towers or other elevated areas that include surface morphology (see, e.g. Meier et al. (2026) for *plume shadows* originated from elevated emitters). Therefore, both the elevation of the plume origin and the surface morphology could be considered if site-specific parallax effects are considered.

This manuscript is connected with the idea of *training as you measure*. That is, the simulated $\Delta X_{\mathrm{CH_4}}$ must be closest possible to the satellite observations. For example, during $U_{\mathrm{eff}}$ calibration, noise is typically added to simulations Varon et al. (2018). However, the angular configuration of the illumination and satellite observation has proven to have a significant impact on the $U_{\mathrm{eff}}$ calibration. The modelling of angles reduces the gap between the simulations and real observations but, still many other parameters can be modelled that include point spread function or stray-light.

The study was carried out at $20 \times 20\,\mathrm{m}^2$ spatial resolution and cannot be directly applicable to large-area mappers. For these missions, the large pixel size might reduce the impact of parallax. However, longer and higher plumes might have a more important impact.

The case of methane simplifies the processing under the assumption of a very low diffuse component of scattered photons in the atmosphere. Its extension to other trace gases such as $CO_2$ might be possible but might not be straightforward for those detected at much lower wavelengths. The retrieval of $NO_2$ is significantly affected by Rayleigh scattering and aerosols, and the AMF can no longer be simplified to a geometric AMF. Therefore, the standard method is to first calculate the slant column density of each pixel, and then convert it to the vertical column density based on a specific AMF (Varon et al., 2024; Borger et al., 2025). A more realistic simulation involves the use of 3D radiative transfers, such as the work in Kushner et al. (2025) for $SO_2$ plumes.

The examples focus on satellite instruments, but these concepts equally apply to airborne or drone detections. In those cases, a lower altitude and pixel size might play a more important role. For example, it might be possible to consider the expected wind direction and solar positions to define an optimised flight path and sensor pointing. Similarly, this could be considered for the design of methane-controlled releases.

Finally, we also expect an impact on the training of detection/quantification models since we have seen how the shape can largely vary. This is directly explained if the training dataset is based on simulations (e.g. Plewa et al. (2025); Jongaramrungruang et al. (2022)). However, even if real plumes are considered as in Vaughan et al. (2024), the angular information could be considered as part of the training and for representativeness (year, latitude and swath representativeness).

*Code and data availability.* The code is available in Gorroño (2026). The data are available in Gorroño et al. (2026)

*Video supplement.* Valverde and Gorroño (2025) displays a mock-up of a methane plume with ground projections for both illumination and observation conditions. The observation is set to nadir and the illumination changes to different azimuth values.

*Author contributions.* JG conceptualize the study and methodology. ZP and JG designed and implemented the code and simulations. AV created a video supplement to the manuscript and analysed the results. JG wrote the original draft, and all authors reviewed and edited the manuscript.

*Competing interests.* The contact author has declared that none of the authors has any competing interests.

*Acknowledgements.* This study was carried out as part of the MEDUSA project (Methane Emissions Detection Using Satellites Assessment) funded by the European Space Agency (ESA Contract No. 4000143908/24/I-LR).

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
