# Peer review of "Considering the observation and illumination angular configuration for an improved detection and quantification of methane emissions"

_EGUsphere, 2025_

## Author Comment (AC1)

We thank the reviewer for the careful reading and constructive comments. Below we respond point-by-point (reviewer comments in blue, our responses in black).

The presented data shows some unexplained behavior, and I am uncertain if the quantitative results are statistically reliable. This may be achieved easily by increasing sample size or showing some additional investigation, but is essential to enable reproducibility. I also recommend publishing the datasets and software.

Thanks for the comments. It is completely correct that we needed to increase the dataset. We made an important effort to increase the dataset from the initial 4 to 11 different geosthrophic winds. With this new dataset the results show much more clear (e.g. DoP is more obvious and no "jumps" appear on the polar plots.

The datasets are now publicly open to the community
https://doi.org/10.5281/zenodo.18161182
The full code is also available here
https://doi.org/10.5281/zenodo.18242174

**Major Comments**

I have two major comments:
1. Your analysis focuses on the IME method as presented by Varon et al., 2018. A major assumption of this method is that it is possible to approximate the plume length by taking the square root of the area. While this ends up with the correct physical dimension, this assumption carries some obvious flaws. Those are somewhat mitigated by the calibration process of $U_{eff}$, as you show in the paper. I wonder if the results of this study can provide insights in the following questions, and if you can elaborate in the manuscript:
   1. If we want to continue using the IME method in its current form, is there a better predictor for the plume length than the square root of its area? E.g., the extent along the wind direction?
   2. Since the observation geometry mostly changes the plume shape, is the current IME method exceptionally prone to biases wit respect to the observation geometry? E.g., is the cross-sectional flux method more robust against this error contribution? For the extreme example in your Figure 2, the

plume area *doubles* while the mass across a transect *stays the same*.

I agree with the reviewer that the consideration of angles triggers an important discussion on the impact on different methodologies.

Alternative definitions of L, such as the plume extent along the wind direction or the maximum plume length, are possible and have been explored in additional sensitivity tests, we have added the following paragraph in the discussion section:

"We also tested an alternative definition of the plume length scale $L$ based on the maximum plume extent after noise addition, rather than the square root of the plume mask area used throughout this study. This alternative formulation results in a redistribution of the angular error patterns, with errors tending to concentrate preferentially in the lateral directions of the polar representation rather than symmetrically on either side. This is expected from a plume length definition less affected by plume shape broadening. Nevertheless, the overall magnitude of the errors and their systematic dependence on the observation and illumination geometry remain consistent. These results indicate that, while the specific definition of L influences the spatial expression of the errors, the dominant driver of the uncertainty is the angular configuration itself, reinforcing the main conclusions of this work."

This is a polar plot result of the Ueff:

[Figure]

Because the objective of this study is to highlight the importance of accounting for observation and illumination geometry, rather than to compare alternative plume length definitions or retrieval methodologies, we only briefly mention this sensitivity in the

Discussion without expanding into a methodological comparison. We agree that explicitly considering angular effects naturally motivates a broader discussion on quantification methods and training strategies. To facilitate further exploration, both the code and the full simulation framework are made openly available, allowing the community to extend these concepts and apply them to instrument- or application-specific scenarios.

2. In L227 you note that you collect 480 sample plumes that lie 30 seconds apart. Since plumes evolve on many time scales, 30 seconds may be too short to de-correlate your samples. This means that your 480 synthetic observations may not be independent and, thus, could lead to biased results. How long do the plumes need to evolve before they become uncorrelated? One hint that correlation may affect in your study is the unexplained feature in the polar plots that you point out in L276. Your plots show unphysical jumps in approximately (exactly??) 120 degree intervals, easily visible in Figure 8.

   1. If this is indeed a result of the sample size, could you provide a plot of the sample size for each data point in the polar plot?
   2. If certain geometries filter out a significant portion of the data, you should investigate why this happens. Are those correlated samples, and the effect vanishes with larger sample size? Or is there a physical reason for this behavior?

We investigated temporal correlation and adjusted our sampling to reduce potential bias. Specifically, we increased the dataset by including more geostrophic wind realizations and selected samples spaced 60 s apart (following the decorrelation guidance cited in the manuscript and defined in Ouerghi et al. (2025)). As a result, the number of independent samples increased (480 to 660) and the previously observed abrupt jumps in the polar plots disappeared. The updated PoD and Ueff polar plots are clearer and consistent with our interpretation that the earlier artifacts were due to limited sample size and temporal correlation.

We attach here the examples for the slope of Ueff and the PoD with no jumps and much more clear directional effects for the latter.

[Figure]

The new figures and results are included in the reviewed version.

**Minor Comments**

- L12: The calibration of flux rates against U$_{eff}$ is unique to this form of the IME among the mass balance approaches. You should mention that you base the results for emission estimation on a this specific method.

Correct, we changed the sentence to " *We tested the impact in the methane flux rate using the integrated mass enhancement (IME) method*."

- L14/15: "These errors are explained..." - the changing plume area does matter most in the IME method. Again, name the method in the abstract.

Changed to:"The errors in the IME method are explained"

- L38: Same as above, Divergence-Integral methods or

Changed to: "In divergence-integral methods,..."

- L63ff: When mentioning physical based retrievals of greenhouse gases, I recommend including Butz et al., 2010 (doi:10.1029/2011GL047888) and O'Dell et al., 2018 (doi:10.5194/amt-11-6539-2018).

These two papers are now included in the references.

- L69: "This reference is generated from a statistical reference background spectrum from the image further convolved with the methane absorption spectrum" - I believe this sentence is inaccurate. My understanding: "The reference is created from spectra in the background observations in the image. The methane absorption spectrum is convolved with the instruments spectral line shape, providing the target signature. The matched filter fits each spectrum using the reference spectrum and the target signature, providing a pixel-specific enhancement." I know

this is longer, but I do not believe that the reference spectrum is ever 'convolved' with the methane absorption spectrum.

Thanks. Your description is correct. We wanted to be concise but it is correct that might need better description. Included your description verbatim.

- L72: "It assumes …" - this sentence is basically impossible to understand without reading Varon et al., 2020, and adds little to the presented work. I suggest cutting it.

We have deleted the sentence.

- L77 / eq. (1): It is unclear what those quantities are. Please make sure to describe them properly. The ΔXCH4 is a *vertical column enhancement in molecules/cm2*. What is lambda?
First paragraph of subsection 2.1 defines ΔXCH4. we have made a small modification to further clarify what lambda means.

- L84: The geometric AMF also assumes a plan-parallel atmosphere. That breaks down at high SZA or VZA, but you never reach them in typical situations with nadir-viewing instruments. May be worth noting, it confused me when I read the manuscript for the first time.

Thanks for pointing at this. Added:
"Furthermore, Equation \ref{eq:amf} also assumes a plane-parallel atmosphere. That might be invalid for very high values of SZA or VZA, but does not typically occur for nadir-viewing instruments."

- L89: "We define parallax …" - this sentence is exactly copied from the introduction. I suggest putting it in where it fits better, not repeating it.

Thanks for pointing at this repeated sentence. We have deleted the one in L89.

- L110: A bit of an understanding question on my part - assuming no noise and artifacts, will the total mass in the plume be conserved? Or only nearly conserved? I would really like one of those polar plots that shows how the total mass in the plume changes with the geometry, if it is not strictly conserved.

This is a very important question. The mass is conserved by definition (we have the same 3D volume mixing ratio). The only change is in the spatial distribution. I have not produced a full polar plot because it is more useful to understand as a table.

| SZA0 SAA0 | SZA30 SAA0 | Diff% | SZA0 SAA0 | SZA30 SAA0 | Diff% |
|---|---|---|---|---|---|
| 377719.1 | 374203.6 | 0.9 | 0.0 | 0.0 | #DIV/0! |
| 385496.2 | 374863.3 | 2.8 | 13479.8 | 13479.8 | 0.0 |
| 388731.6 | 366524.6 | 5.9 | 24798.7 | 24798.7 | 0.0 |
| 375886.8 | 347024.4 | 8.0 | 36012.7 | 36012.7 | 0.0 |
| 358056.2 | 326314.4 | 9.3 | 46756.3 | 46756.3 | 0.0 |
| 344218.3 | 313239.8 | 9.4 | 56701.7 | 56701.7 | 0.0 |
| 333182.0 | 306731.8 | 8.3 | 65964.0 | 65964.0 | 0.0 |
| 321542.7 | 302124.2 | 6.2 | 75801.5 | 75801.5 | 0.0 |
| 320855.7 | 301703.6 | 6.2 | 86291.9 | 86291.9 | 0.0 |
| 331704.3 | 308500.4 | 7.2 | 97676.3 | 97676.3 | 0.0 |
| 335158.0 | 309107.7 | 8.1 | 107430.0 | 107430.0 | 0.0 |
| 330800.0 | 304631.2 | 8.2 | 114324.8 | 114324.8 | 0.0 |
| 323111.4 | 300242.1 | 7.3 | 121575.6 | 121575.6 | 0.0 |
| 309875.5 | 294133.7 | 5.2 | 131107.0 | 131107.0 | 0.0 |
| 298501.8 | 289901.5 | 2.9 | 140564.4 | 140564.4 | 0.0 |
| 294832.0 | 286585.9 | 2.8 | 153136.5 | 153136.5 | 0.0 |
| 307419.1 | 295543.6 | 3.9 | 165248.4 | 165248.4 | 0.0 |
| 323572.5 | 305382.0 | 5.8 | 173452.1 | 173452.1 | 0.0 |
| 329564.9 | 308365.1 | 6.6 | 178227.8 | 178227.8 | 0.0 |
| 334303.7 | 307431.5 | 8.4 | 183304.9 | 183304.9 | 0.0 |
| 329184.6 | 297401.4 | 10.1 | 195704.7 | 195704.7 | 0.0 |
| 305806.6 | 278492.3 | 9.3 | 216409.4 | 216409.4 | 0.0 |
| 284959.1 | 263623.3 | 7.8 | 231011.2 | 231011.0 | 0.0 |
| 276221.6 | 260855.2 | 5.7 | 238851.9 | 238845.2 | 0.0 |
| 277028.8 | 264520.2 | 4.6 | 249005.8 | 248971.3 | 0.0 |
| 287214.9 | 275381.3 | 4.2 | 262244.6 | 261957.2 | 0.1 |
| 301205.2 | 288626.3 | 4.3 | 274957.4 | 272915.4 | 0.7 |
| 308745.1 | 295295.6 | 4.5 | 285184.6 | 280092.8 | 1.8 |
| 309209.0 | 294034.3 | 5.0 | 293747.6 | 285105.9 | 3.0 |
| 302465.0 | 284149.4 | 6.2 | 302727.8 | 291641.7 | 3.7 |
| | | | | | |
| 2m/s multisource range(60,120,2) | | | 2m/s multisource range(0,30,1) | | |

The table just took a simulation dataset with a geostrophic wind of 2m/s and calculated the mass (with no added noise) for the snapshots from 30min to 1hour every minute. The result (left three columns) shows clear underestimation of the mass after angular consideration.

This is not a consequence of changing mass but of cropping the ROI after angular shift (higher altitude layers can reach large shifts). The right columns show the same mass for the first 30 samples (15min simulation). Only when the plume is long enough and reaches the end of the ROI, there is a small difference of mass.

To sum up, the mass is conserved but the ROI size and large shifts and high altitudes are slightly outside and cropped. This is not concerning since our approach masks the plume after adding the noise.

- L123: There is a significant difference between those works when it comes to IME. Frankenberg uses something more akin to the cross-sectional flux, while Varon introduces the effective wind speed and the sqrt(A) length scale that this paper employs.

Thanks for clarifying. We have slightly change the paragraph to point at this.

- L135: Why the underestimation?

I referred to the underestimation described in L107 "AMFg is overestimated because only the downward or upward paths must be considered for each pixel. Then, the impact on ΔXCH4 is inversely proportional to the AMFg attending to equation 2.". It reinforces the idea that, although the error is cancelled at IME, it exists at the enhancement map level. I agree that it can be a little bit confusing and the sentence has been deleted.

- L191 / eq (11): This equality is a bit weird. Did you arrive there by comparing the term to the exponent of eq (9)? Make sure the math is easy to follow, an additional line wouldn't hurt here.

We compared exponents in eq 9. To clarify that, we have introduced eq 11 and 12 to explain better the equivalence.

- Figure 3: Please add x, y, and z axis to the plots.

The axis are now included and the caption includes: "A small axis glyph is included in each panel to indicate the along-plume (x), cross-wind (y), and vertical (z) directions."

- L221: Why is that interesting? Validating your modeling scheme? Or is it a mere coincidence?

We have slightly rephrased as:
"Finally, for the same case (SZA 40°, VZA 40°), the plume at RAA -90° represents a nearly symmetric double plume very similar in shape and angular configuration to the WV3 plume presented in Figure 2." We want to point out that for similar angular configurations, we can simulate the satellite observed plumes. In a way, there is a correspondence between real observation and our simulations which can be understood as a validation.

- Figure 6: Please state explicitly that the angular axis is the SAA and radial axis is the VZA.

We have included in the caption of Figures 6, 7 and 8, the following statement:
"The angular axis represents the RAA angles and radial axis refers to the SZA angles."

- L259: "..., the errors ..." - is this a random error or a bias? Your analysis points toward a bias, which is far more concerning than an additional error term.

It is a systematic error or bias for the specific illumination angle. We reproduce the same training for each illumination angle and a correction bias should be applied. We bring that in the discussion and propose that this is considered in the quantification, validation or learning algorithms.

- L270: I guess it comes out the same, but decreasing the flow rate would be more realistic than 200 ppb noise.

You are correct but we rerun the experiment increasing to 300ppb (we need a full day in the server to reprocess). We have included a small comment clarifying that reducing the flow rate is analogous.

- L279: Decreasing PoD - That makes sense - PoD at high speeds is generally lower since the methane is transported quickly and enhancements are smaller.
You can provide this as an explanation to the observation.

Due to the increased sample size and reduced sample correlation (as explained before), our results for PoD are clearer.

We have slightly increased to 300ppm noise and PoD results in values from 0.5 to 0.3. We split the results for high vs low winds (3m/s) and the PoD ratio (against nadir case) shows small changes at low wind. However, for larger winds the PoD can be double from nadir and 4x from peak-to-peak.

[Figure]

- L319: Please add a discussion of the potential of alternative point source estimation methods.

We have improved the discussion section with the followinf text: "Explicitly considering angular effects naturally motivates a broader discussion of quantification methods and training strategies. For example, it is possible that other methods, such as the Cross-sectional Flux, are more robust against the parallax effect since no area is required. As mentioned above, we have released the code and simulated data in \citet{gorrono2026_CH4unc} and \citet{gorrono2026_benchmarksim} to facilitate the application to other scenarios."

- L341: state that 'diffuse' refers to the scattered photons. It could

also be atmospheric transport or a type of methane sources in the context of this publication.

Thanks, we have clarified that point adding: "...low diffuse component due to scattered photons in the atmosphere."

**Technical Comments**

- L101: "reprocessed" strikes me as a bit of a weird wording choice, or

did you re-create the whole evaluation for this paper. I suggest phrasing it like "Figure 2 shows a double plume as presented in a case study by Sánchez-García et al. 2022)".

- L147 / eq (5): This is probably part of the typesetting, but make

sure you use consistent indexing. Only indexes are italic, shorts like "eff" and "ref" should be normal font.

- L162: "resultingThe"

- Figure 4: Please use consistent fonts and font sizes

- L228: Representative for the whole manuscript, please use a

half-space (\,) between numbers and units.

- L260: 10% to 30%

- L270: possibility —> hypothesis

- L292: "a lot of compute resources" is colloquial English. Suggest "Is

computationally expensive"

- L312: "inThe"

Thanks for these technical suggestions. We have considered them and made necessary changes in the corrected version

---

## Author Comment (AC2)

We thank the reviewer for the careful reading and constructive comments. Below we respond point-by-point (reviewer comments in blue, our responses in black).

1. While the impact of the observation and illumination angular configuration on quantification is analyzed comprehensively, the impact on retrieval is minorly discussed only when reviewing the mathematical expression. I would recommend underlining just the impact on quantification throughout the manuscript. One example would be revising line 4-5.

It is a very good point and needs to clarify that 1) we describe the effect in methane fields and 2) quantify its impact on flux rates.

*Abstract: clarified one sentence*
This paper describes how observation and illumination angular configuration affects satellite-derived methane enhancement fields and, more critically, the calibration and uncertainty of semi-empirical emission rate estimation models, using simulated datasets.

*Introduction: introduced one sentence*
The analysis of $\Delta X_{\mathrm{CH}_{4}}$ maps is used to illustrate physical mechanisms, whereas the quantitative assessment focusses on emission flux calibration, the uncertainty, and the probability of detection.

*Section 3.1: add a sentence*
In this subsection, the impact on $\Delta X_\mathrm{CH_{4}}$ maps is analysed to illustrate changes in plume shape, without attempting a quantitative retrieval performance assessment.

*Discussion: add a sentence*
Consequently, uncertainties in $\Delta X_\mathrm{CH_{4}}$ are not evaluated independently, but through their propagation to flux quantification.

2. It would be nice to have "(SZA=0)" and "(VZA=0)" in line 93 connecting the term "observation and illumination angle" to SZA and VZA. Furthermore, it would be nice to have SZA and VZA illustrated in Figure 1.

Thanks. All the changes are now included in the manuscript.

3.  How is L the plume length defined? Are you following Varon et al., 2018? Please provide a description. Also, is there a better way to define L? What about calculating the length along the wind direction or the farthest length?

We thank the reviewer for raising this point. In this study, the plume length scale L is explicitly defined as the square root of the plume mask area, following the standard implementation of the IME framework in Section 2.1 when describing Eq. (4).
We have clarified that sentence by rephrasing as:

"L the plume length scale in m, which is typically approximated by the square root of the plume mask area, **following the standard IME formulation of Varon et al. (2018).**"

Alternative definitions of L, such as the plume extent along the wind direction or the maximum plume length, are possible and have been explored in additional sensitivity tests, we have added the following paragrapgh in the discussion section:

"We also tested an alternative definition of the plume length scale $L$ based on the maximum plume extent after noise addition, rather than the square root of the plume mask area used throughout this study. This alternative formulation results in a redistribution of the angular error patterns, with errors tending to concentrate preferentially in the lateral directions of the polar representation rather than symmetrically on either side. This is expected from a plume length definition less affected by plume shape broadening. Nevertheless, the overall magnitude of the errors and their systematic dependence on the observation and illumination geometry remain consistent. These results indicate that, while the specific definition of L influences the spatial expression of the errors, the dominant driver of the uncertainty is the angular configuration itself, reinforcing the main conclusions of this work."
This is a polar plot result of the Ueff:

[Figure]

Because the objective of this study is to highlight the importance of accounting for observation and illumination geometry, rather than to compare alternative plume length definitions, we only briefly mention this sensitivity in the Discussion without expanding into a methodological comparison. We agree that explicitly considering angular effects naturally motivates a broader discussion on quantification methods and training strategies. To facilitate further exploration, both the code and the full simulation framework are made openly available, allowing the community to extend these concepts and apply them to instrument- or application-specific scenarios.

**4.** I don't see any use of equation 7 and 8. What are you trying to explain with these?

I agree that they are not directly mentioned further in the manuscript, but clarify the conversion of 3D volume mixing ratios in to 2D enhancement maps.

Since we have now published the code, I directly refer to it and make that connection in the manuscript.

**5.** I believe AA in equation 8 is azimuth angle. Please define AA in the text.

We have now clarified stating "where $AA$ refers to the azimuth of the viewing or solar angles."

**6.** I think equation 10 should be underlined to highlight that the authors are varying SZA and VZA to analyze how the quantification is affected.

We agree and have clarified the role of Equation (10) by explicitly stating that SZA and VZA are systematically varied in the analysis to assess how angular configuration propagates into emission flux quantification. Explicit sentence:

"Equation \ref{xch4updowncombination} highlights the explicit dependence of the geometric air mass factor on SZA and VZA, which are systematically varied in this study to assess their impact on methane flux quantification."

**7. Please explain how Figure 3b and 3c was generated. x and y labeling might help. Is y-axis in figure 3b elevation?**

The axis are now included and the caption includes: "A small axis glyph is included in each panel to indicate the along-plume (x), cross-wind (y), and vertical (z) directions."
To better clarify how they are obtained, we have added a short sentence: "The maps are calculated collapsing the across and along the plume directions rather than the vertical one."

**8. Figure 6: a detailed description of the polar plot is needed using RAA and SZA.**

We have included in the caption of Figures 6, 7 and 8, the following statement:
"The angular axis represents the RAA angles and radial axis refers to the SZA angles."

**9. Before explaining Figure 7, please describe that using Ueff simply defined from vertical integration (SZA=0 and VZA=0) of WRF-LES model would cause error, and you are showing the how big that error would be using a ratio.**

We have now better explained Fig 7 with the following text:
"Defining the values of Ueff based on $\Delta$XCH4 maps from the vertical integration (SZA=0 and VZA=0) of WRF-LES model270
results in errors. Figure 7 normalises Ueff against the vertical integration scenario (SZA=0 and VZA=0) to show the relative error in the polar coordinates for different U10 winds."

Minor Comments:

- Figure 1: Dx should be Dd to match the text in the manuscript. In the text, downwelling path (before reflection) and upwelling path (after reflection) is described to be green and red respectively, but the figure doesn't match this description.
- Line 162: remove "resulting" before "The resulting"
- Figure 5b: change SZA 0 to SZA 60
- Line 239: change SAA to RAA.
- Figure 7: change SAA to RAA.
- Line 312: remove "in" before "The results"

Thanks for these minor suggestions. We have considered them and made necessary changes in the corrected version

---

## Referee Report (RR1)

The authors have taken into account all of my previous comments in a thoughtful way. I would recommend publication after reflecting minor comments below.

Minor Comments:

- Line 43-47: Divergence–integral method is an alternative to the IME method for quantification. Do you mean some other retrieval method?

- Line 64: replace DoP with PoD

- Line 79-80: "The methane absorption spectrum is convolved with the instrument spectral line shape and then multiplied with reference spectrum, providing the target signature." is a more correct description of matched-filter method.

- Figure 1: I suggest using solid line and dashed line along with the different colors for easier separation of downwelling and upwelling path that is described in Line 105.

- Line 139: I believe $k$ is derived assuming standard temperature and pressure. Please provide the details.

- Line 152: I would suggest describing L as defined during emission quantification instead of retrieval.

- Line 204: Do you mean Equation 10?

- Use "Ueff calibration" instead of "calibration" throughout the manuscript.

---

## Author Response (AR2)

Thanks for pointing at this minor changes that will help us improve the manuscript. Here below, we provide answer and changes in the document.

Minor Comments:
- Line 43-47: Divergence–integral method is an alternative to the IME method for quantification.Do you mean some other retrieval method?

Thanks for pointing at this. We have pointed specifically at the IME method.

- Line 64: replace DoP with PoD

Thanks, it has been changed.

- Line 79-80: "The methane absorption spectrum is convolved with the instrument spectral line shape and then multiplied with reference spectrum, providing the target signature." is a more correct description of matched-filter method.

Thanks, it has been changed.

- Figure 1: I suggest using solid line and dashed line along with the different colors for easier separation of downwelling and upwelling path that is described in Line 105.

We have changed the figure and line 105.

- Line 139: I believe k is derived assuming standard temperature and pressure. Please provide the details.

k is explicitly described in the code as:
ime=(8000* np.sum(xch4[plume] / 1000)* ueff_config.pix_res[0]* ueff_config.pix_res[1]* 1000* 0.01604/ (1e6 * 22.4))
we have been more explicit in the text rewriting as:
"k is a scaling factor that converts the total of the pixel-wise methane concentration values in ppb to kg by assuming Avogadro's law, the molar mass of methane (0.01604 kg/mol), an atmospheric column height of 8000 m and taking into account the pixel size (e.g. k =5.155·10−3 kg/ppb for 30 m pixel)."
The assumption of a standard temperature and pressure is taken into account when converting the 3D VMR into a 2D concentrations described in Equation 6. In this same conversion, the term 8000 is also introduced and, therefore, is cancelled when converted into flux rates in equation 5.

We preferred not to expand the discussion to other areas in this manuscript but rather distribute the code and simulations to the community.

- Line 152: I would suggest describing L as defined during emission quantification instead

of retrieval.

L is part of the process between retrieval and quantification. We have rewritten as: "However, its impact is based on the definition of both L in equation 4 and Lref during Ueff calibration." to keep it neutral.

- Line 204: Do you mean Equation 10?

Thanks. It has been changed.

- Use "Ueff calibration" instead of "calibration" throughout the manuscript.

Thanks, we have identified two instances that required specify the type of calibration.